# Nowcasting Entrepreneurship: Urban Third Place versus the Creative Class

Li Fang [1,*] and Timothy Slaper [2]

1. Department of Urban and Regional Planning, Florida State University, Tallahassee, FL 32308, USA
2. Kelley School of Business, Indiana University, Bloomington, IN 47405, USA; tslaper@indiana.edu
* Correspondence: lfang3@fsu.edu; Tel.: +1-850-644-4512

**Abstract:** Researchers have long debated whether entrepreneurship policy should focus on place or people. In this paper, we extend the place-based versus people-based theories using contemporaneous and geographically granular web-user online activity data to predict a region's proclivity for entrepreneurship. We compare two theoretical hypotheses: the urban third places—informal gathering locations—that facilitate social interaction and entrepreneurship, in contrast to the creative class which fosters entrepreneurial energy and opportunity in a region. Specifically, we assess whether business formation has a stronger statistical relationship with the browsing behavior of individuals visiting websites associated with third place locations—e.g., restaurants or bars—or the concentration of web browsing behavior associated with "the creative class". Using U.S. county-level data, we find that both urban third places and the creative class can predict about 70% of the variations in regional business formation, with the creative class having a slight competitive edge.

**Keywords:** entrepreneurship; website browsing behavior; nowcasting; urban third place; creative class

## 1. Introduction

Researchers have long debated whether entrepreneurship policy should focus on place or people [1]. The place-based approach emphasizes improving the built environment to spur social interactions and exchange ideas [2]. In contrast, the people-based approach focuses on cultivating human capital [3]. Both approaches have been adopted in practice in American cities [4].

The competing theories of urban third places and the creative class are a reflection of that debate. On the one hand, urban third places—informal gathering and mixing locations [5] such as restaurants, coffee shops, and bars—encourage social interaction, which breeds new ideas and subsequently leads to innovation and entrepreneurship. Thus, an urban area abundance of third places is expected to be more entrepreneurial. On the other hand, the creative class theory emphasizes people as the driver of regional entrepreneurship. Florida defined the creative class as scientists and engineers, university professors, poets and novelists, artists, entertainers, actors, designers, and architects, as well as the "thought leadership" of modern society [6]. He argued that the ability to attract or cultivate the creative class is key for cities to form new businesses.

These two theories, though different, are not mutually exclusive. For example, urban third places may attract the creative class; the creative class may also encourage the formation of welcoming urban third places [6,7]. That said, creative class theory implies the effectiveness of education policies such as art schools and designer workshops do not necessarily follow naturally from the urban third place theory. Thus, statistically comparing these two theories is important for both academic and policy-making purposes.

However, a gap persists in the understanding of the empirical outcomes of these two theories. Research has tested these two theories in isolation, but not have compared them using similar data and methods. Moreover, the existing explanatory power of the

empirical evidence is mixed. For example, Cushing found that measures of talent, diversity, and creativity explain a great deal of regional growth [8]. In contrast, Donegan, Drucker, Goldstein, Lowe, and Malizia did not find that the creative class indicators explain the metropolitan job and income growth any better than traditional indicators [9]. Cabras and Mount, using pubs as an example, showed the positive impact of third places on local economies, e.g., employment and business opportunities [10]. Similarly, Grodach examined how small art spaces contribute to building social networks and advancing the art economy [10]. Despite the aforementioned ground-breaking research on third places and the creative class, there is a dearth of research comparing the explanatory power of Who versus What i there is, in terms of the regional benefits and performance of regions.

This comparison is also important for entrepreneurship policies. Understanding which matters more can inform policy initiatives. Based on evidence, policymakers can make decisions to focus their resources towards either improving their built environment or cultivating and retaining creative human capital through educational programs, training, and/or the development of a tolerant culture. Given that many traditional economic development policies—tax abatements or subsidies for example—are only marginally effective at best [11], novel policy tools are needed. Many economic development catalysts adopt a mixed approach that embrace elements of both the people-based and place-based approaches [4], but whether that mixture can be considered the most effective is still unanswered.

We fill this knowledge and policy making gap by competing the urban third place and the creative class theories, that is, putting the two theories and empirical results in a horse race to test which better predicts regional entrepreneurial activities. Specifically, we collected contemporaneous and geographically granular user online web behavior to estimate interest in urban third places in contrast to people revealed preferences as members of the creative class. We find that both can effectively explain about 70% of the variation in entrepreneurial activities. The creative class indicators have a small competitive edge in their predictive power.

Our contribution is four-fold. First, we conduct a formal comparison between these two theories, which helps to understand their empirical power. More broadly, this analysis contributes to the debate about the people-based versus place-based planning approaches. Second, this paper introduces a new data source—users' online browsing behavior—to the study of entrepreneurship regionally. In the information age, user-generated online data are abundant and in real time. The ability to harness these data to test spatial and entrepreneurial theories can take the theory-building and testing to the new level of precision; precision that is fast-adapting and registers phenomena in almost real time. Third, this paper shows that users' online browsing profiles are a better measurement for the creative class than the traditional measurement using occupational data. Creativity is not necessarily tied to one's job. There are avocations. Our measure defines people as the creative class based on online behaviors rather than their jobs, which better captures their "creative class" identities. Finally, we provide evidence-based practical strategies for local planners and other policymakers to help them cultivate a more entrepreneurial culture and encourage the creation of new businesses and economic vitality.

## 2. Literature

### 2.1. The Determinants of Urban Entrepreneurship

What makes a city entrepreneurial? Broadly speaking, three categories of factors are found, including institutional arrangements, the regional and urban built environment, and the types of firms and population characteristics. Since North [12] and Williamson [13], the importance of institutions on economic performances has been well documented. Building proper institutions that minimize transaction costs and protect the incentives for innovation and entrepreneurship are key to economic prosperity [14].

At the regional and city-level, the business ecosystem and built environment significantly affect entrepreneurial activities [15]. The presence of existing entrepreneurs and entrepreneurial culture can draw more entrepreneurs into the picture [16]. As a result, en-

trepreneurship is highly clustered in global cities and metropolitan areas, and even within certain city districts. Renski found that in the United States, suburbs, small cities, and rural segments of the metropolitan areas often have relatively high rates of new firm entry and survival [17]. Both localization economies [18]—industry clusters that share similar supply chains and labor force requirement and urbanization [19]—develop collections of diverse industries that are conducive to firm births and growth [20,21]. Andersson, Larsson, and Johannisson highlighted the point that social interactions among diverse populations can drive business formation and growth [22,23].

The urban built environment also plays a role. For example, Smit found that the visual quality of a district is critical for the locational choices of creative entrepreneurs [24]. By extension, transit and transit-oriented development are also associated with creativity and the start of businesses [25].

Lastly, the type of firm and individual characteristics of the business founder also matter. Small and large firms have divergent effects on entrepreneurship [26]. Networking capacity [27], social skills [28], and having a balance of diverse skill sets [29] all affect the probability of successfully starting a new business.

### 2.2. Urban Third Place

The theory of urban third place spins off from the discussions on how an urban built environment encourages social interactions. Urban third places, defined as informal gathering locations outside home and work [5]—restaurants, coffee shops, bars, libraries, and parks for example—are an important component of the urban built environment. These places are less formal and restrictive compared to workplaces, but still offer local public or semi-public spaces to safely engage with strangers. As a result, these places fill a gap between the private and the public as well as the internal and the external. These places provide a unique experience only available in an urban setting [30]. They foster the formation of social ties and help people exchange ideas and discover opportunities [10,31].

Urban third places provide social and economic benefits. For example, Cabras and Mount found that pubs bring communities together and promote economic development opportunities [10]. Specialty coffee shops also provide customers a unique "third place" urban experience, which characterizes togetherness and belonging [32]. Bookman and Waxman found that these coffee places can spur social interactions, construct hospitable space, and establish on-going social relations [32,33]. Peters, Elands, and Buijs found that parks facilitate social interaction and cohesion [34]. Grodach also found that art spaces function as a conduit for building social networks, which subsequently contribute to community revitalization and the development of artists [10]. Similarly, open-air art performance venues [35] and festival spaces [36] are also third places that contribute to relationship building and economic growth. The accessibility to urban third places is found to be positively associated with the perceived quality of life [37].

However, most studies focus on the social benefits of urban third places, while the economic benefits are less frequently studied, except a few mentioned above. Research that explicitly measures the relationship between urban third places and entrepreneurship is lacking. Given that urban third places facilitate social interaction and relationship building, which are important ingredients of entrepreneurship, the impact of these places on establishment births warrants detailed examination.

### 2.3. The Creative Class

The theory of the creative class is rooted in the theory of human capital and urban diversity [6,19,38]. The creative class includes scientists and engineers, university professors, poets and novelists, artists, entertainers, actors, designers, and architects, as well as the "thought leadership" of modern society: nonfiction writers, editors, cultural figures, think-tank researchers, analysts, and other opinion-makers [6]. These are people with creative and critical thinking. "Think creatively" is defined by O*NET as "'developing, designing or creating new applications, ideas, relationships, systems or products, includ-

ing artistic contributions." Thus, these people are able to embrace and engage with new ideas, and actively come up with novel thoughts, methods/applications and products at the frontier of science, technology, and/or arts. At the same time, these people are also absorbing information with a pair of critical eyes; they exercise independent, logical, multi-perspective, and multi-dimensional thinking, express their opinions, and engage in public debates to mark their influence and become the "thought leaders" of the society. The creative class is key to urban economic success. The creative class is correlated with innovation and high-tech industry growth, and with growth in regional employment and population [6].

Empirical results are mixed. Florida, Mellander, and Stolarick and Gabe found that the creative class is strongly associated with higher wages [39,40]. McGranahan and Wojan found that the creative class can predict regional employment growth [41]. Boschma and Fritsch identified a positive relationship between the creative class occupations and employment growth and entrepreneurship [42]. More still, Marrocu and Paci found that the creative class can explain production efficiency [43]. That said, Glaeser found that traditional human capital dimensions, i.e., schooling, outperforms the creative class indicators in explaining urban growth [38]. Similarly, Donegan, Drucker, Goldstein, Lowe, and Malizia found that the creative class indicators do not explain metropolitan job and income growth better than traditional measurements [9].

The policy implications of the creative class theory depart from those of the traditional economic development strategies and the importance of physical infrastructure. Florida argued that the creative class resists traditional economic incentives and, as a result, cannot be attracted to places offering tax breaks or job-training subsidies, sports venues, road capacity and conditions, urban malls, and theme parks [6]. Rather, the creative class is attracted to millennial personal priorities or values—quality experiences, openness to diversity, experimenting with identity, and exploring creativity [6].

Two research gaps persist. First, the empirical results are mixed, which is partly due to the fact that occupation-based measurements for the creative class fall short of precision [6]. The creative class may or may not exhibit their creativity in their jobs. As a result, occupation-based measurements may not capture "the creativity" of the person inhabiting their jobs. Second, the relationship between the theory of the creative class and that of the urban third place has not been well understood. So the question continues unconvincingly addressed: Which better predicts entrepreneurship? Do the two theories and practical outworking complement each other?

## 3. Materials and Methods

### 3.1. Materials

We operationalize entrepreneurship as the count of establishment births by county, reported by Census Bureau Business Dynamics Statistics in 2018. The dependent variables are digital, and geographic snapshots of user web behavior were collected over a day or week in order to capture user interest in and commitment to the creative class and the urban third places at a county-level unit of geographic analysis. The data were initially collected and provided by Dstillery, llc. based on zip code geographic boundaries and aggregated to the Bureau of Economic Analysis county boundaries. In this paper, we have used the web browsing data averaged in June 2020. We acknowledge the time gap between the explanatory and dependent variables. That said, the establishment birth data is the most current publicly available, and the local and regional culture is expected to be stable in the short to medium term [44]. Thus, given the slow rate of demographic and cultural change, one would expect that entrepreneurial regions in 2018 would also be entrepreneurial into the earlier 2020s.

Prior studies typically use the Bureau of Labor Statistics Occupational Employment Statistics data to measure the creative class [40,45]. Approaches have been developed, such as scoring occupations for their level of creativity [46] and using the information on the skill and knowledge requirements of various occupations from the Bureau of Labor

Statistics' Occupational Information Network [41]. That said, creativity at the office, or the lack personal reward for inventiveness at the office, may not capture whether one is creative as an avocation [6].

Our approach captures a user's web browsing activities, and based on their browsing preferences, categorize them into different audiences, or consumer profiles. These data, aside from their original business purposes as consumer profiles and interests, also show the regional, or place-based, concentration of users that exhibit interests in creative class categories. Specifically, we have collected profiles on 37 creative class categories, following the definition of "the creative class" by Florida [6] as closely as possible. To wit, the creative class categories are: Science & Technology, Civil Engineers, Cybersecurity Researchers, Medical Science Researchers, Conservative Think Tank Researchers, Liberal Think Tank Researchers, Data Analysis and Scripting, Science Education & Academic, College Professors, University Research, Poetry Readers, Poetry Fans, Arts & Crafts, Art News & Products, Drawing & Animation, Drawing Enthusiasts, Painting & Renovation, Painting Hobbyists, Photography Interest, Authors, Writing & Publishing, Writing Tools & Citation, Entertainment Industry Decision Makers, Film Production, Memes & Comedy, Humor & Entertainment—Comic Culture, Music Concerts, Live Music, Musical Instrument Purchasers, Architects, Landscape Architects, Commercial Architects, Commercial Contractors & Designers, Design Software, Graphic Design, Healthcare Thought Leaders, and Public Policy Media.

For third places, the categories focus on people's interests as reflected by their web browsing activities, in contrast to some prior studies that focused on the presence of these places [31,36]. This implies that the availability of an urban third place in some proximate geographic locale is not enough. Attracting people's attention and interest is critical and is a necessary condition for converting into the presence of a person (or person's network) at the venue of interest. A wide range of places can be categorized into third places, and in this paper, we focus on the most common types—restaurants, coffee shops, and bars. We have collected data for the top ten restaurants, coffee shops, and bars based on 2018 sales in the nation [47]. Among these thirty brands, all ten restaurants, are included in the Dstillery dataset, and therefore included in this study, but only five coffee shops and bars are available. Thus, the final included data for all the potential third places are 20 brands, and they are driven by Dstillery market and client interest, not researcher preferences. The brands included in our final analysis are ten restaurants, McDonald's, Subway, Taco Bell, Chick-fil-A, Burger King, Wendy's, Domino's, Panera Bread, Pizza Hut, and Chipotle Mexican Grill; five coffee shops, Starbucks, Dunkin Donuts, Caribou Coffee, Peet's Coffee & Tea, and The Coffee Bean & Tea Leaf; and five bars, Buffalo Wild Wings, Chili's Grill & Bar, Hooters, Dave & Buster's, and Beef 'O' Brady's.

These web user-generated data with geographical location have several advantages. They are up to date, and reflect users' interests [48]—thus more accurately identify who is and who is not the creative class based on their lifestyle. The measurement of users' interest in urban third places also helps to tease out irrelevant and unpopular third places. In recent years, there is an increasing trend to harness these type of user-generated data to nowcast economic activities before official government statistics [49]. In addition, using "unconventional data"—data that is not reported as official federal statistics but rather collected as digital vapor trails resulting from electronic device activity—is becoming more conventional [50,51].

Other control variables such as county population, racial composition, and median household income, come from the Census Bureau and aggregated to the Bureau of Economic Analysis county definition that recombines county seats and "independent cities" with their more rural ring areas. The specific variables included are detailed in Appendix A Table A1. There are 3069 counties with complete data for all variables upon which we performed a model training procedure to use the explanatory and control variables to predict establishment births and compare the predictive power between different models.

*3.2. Methods*

We have adopted a machine learning model training and testing procedure to compare the predictive power of urban third place versus the creative class indicators on entrepreneurship. This exercise helps to figure out which theory—the urban third place or the create class—is the major explanatory factor of entrepreneurial activities. The machine-learning procedure is one of the commonly adopted methods in the field of data science, and it fits our research aims for two reasons: (1) applying user-generated data without a pre-defined construct that combines user concentrations across third place brands and creative class categories, in the case of which a machine learning model training method can inform which indicators to include; and (2) using user-generated data to predict entrepreneurial activities, for which machine learning method outperforms traditional statistical methods [52].

To implement the machine learning procedure, we randomly split the whole sample into two subsamples: the training set, which contains 90% of the sample, and the testing set, which contains the other 10%. The training set is used to train the model in order to find the best way to aggregate the user concentrations for third place locations and creative class categories. Within the training set, 90% of the samples are used to train the initial model. The other 10% are used as a validation set to fine-tune the model. This is standard procedure in machine learning to avoid overfitting. The testing set, which contains 10% of the original sample, is used to calculate the predictive power of the trained models. We have repeated the above randomization process a hundred times to avoid non-robust results coming from a single random split.

The two models are specified as follows.

$$y_i = \alpha_0 + \boldsymbol{\alpha_1 TP_i} + \boldsymbol{\alpha_2 X_i} + \varepsilon_i$$

$$y_i = \beta_0 + \boldsymbol{\beta_1 CC_i} + \boldsymbol{\beta_2 X_i} + \varepsilon_i$$

where $y_i$ represents the number of establishment births in county $i$. $\boldsymbol{TP_i}$ and $\boldsymbol{CC_i}$ denote the concentration of third places and the creative class in county $i$, measured by a series of concentration indices that calculate the number of site visits to a particular (type) website per person. The specific forms of aggregation of these concentration indices are chosen through the data-driven model training approach using the Lasso penalty method and a selection criterion of minimizing the mean squared error. The Lasso penalty method is one of the most commonly adopted machine learning method that optimizes the inclusion of a large number of variables to best fit the data while avoids overfitting [53]. $\boldsymbol{X_i}$ denotes other control variables, including county demographic, economic, and social characteristics. Detailed variable descriptions and their sources are listed in Table A1. $\varepsilon_i$ represents the random errors.

Two statistics are in focus. First, we aim to compare the $R^2$ of these two models to establish the better of the predictive powers of urban third place against that of the creative class indicators. This achieves the main research objective of this paper to compare the explanatory power of these two theories. Second, we will also compare the coefficients for different third place brands, i.e., different coefficients within the vector $\boldsymbol{\alpha_1}$, and those for different creative class categories, i.e., different coefficients within the vector $\boldsymbol{\beta_1}$. We have standardized all third place and creative class indicates before using them to train the models, and, therefore, their coefficients can be directly compared. This comparison reveals, within third places or the creative class, which brands and categories are the most predictive of entrepreneurship. While we do not claim any causal relationship here, the high predictive power at least points to a direction worthy of further investigation and potential policy recommendations.

## 4. Results

### 4.1. Geographical Concentrations of Urban Third Places and the Creative Class

The geographical distributions of the creative class and user interests in third places exhibit quite distinctive features, as visualized in Figures 1 and 2. User interests in major restaurants, coffee shops and bars are concentrated mostly in California, Florida, Minnesota, North Dakota, and Alabama. The concentrations in California and Florida are expected, as both states are famous tourist destinations with a somewhat leisurely lifestyle. Those in Minnesota, North Dakota, and Alabama are not as much expected. It is noteworthy that the Dstillery indicators for web browsing activities are scaled to population. As a result, less populated regions may show up as relatively high concentrations of users' interests in third places. This may explain why some states not expected to have high (absolute) concentrations of third places show up as high (relative to population) concentrations.

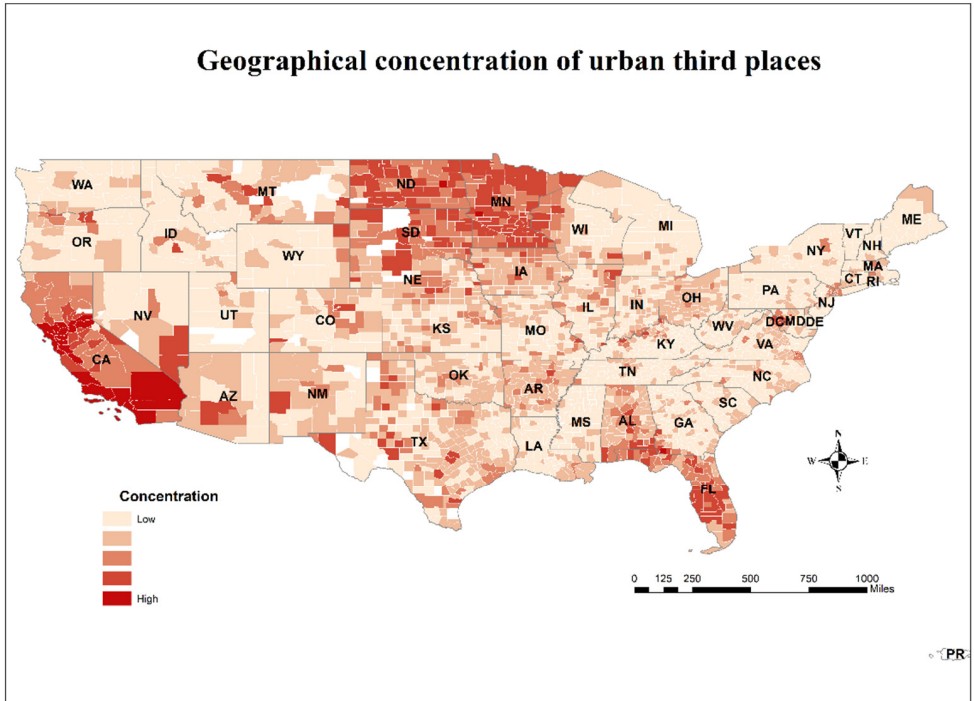

**Figure 1.** The geographical concentration of users' interests in urban third places. Note: (1) This graph visualizes an average concentration index of all third place brands collected. (2) Since the absolute magnitude of the index does not have any meaning attached to it, we only show the relative low and high status of different counties in the graph. The five categories of concentrations are formed by equal intervals.

In contrast, the creative class is more evenly distributed across space, as shown in Figure 2, and relatively more concentrated in the states of Washington, Oregon, southern California, Vermont, Massachusetts, and New Hampshire. Among these major concentrations, Washington, southern California and Massachusetts are states known with high-tech industries and skilled labor. These distinctive distributions enable us to statistically separate the impact of the third places versus the creative class.

The geographical distribution of establishment births is shown in Figure 3. The vast majority of counties in the nation has limited entrepreneurial activities as measured by establishment births, with fewer than ten new business formed in 2018. However, there are a few high entrepreneurial counties with more than 150 startups, mainly located in California, mid- and southern Florida, Massachusetts, Connecticut, New Jersey, Washington, Oregon, Arizona, and Colorado. Taking the three graphs together, both the geographical distributions of third places and the creative class appear to correlate to that of establishment births, but the creative class exhibits a closer relationship.

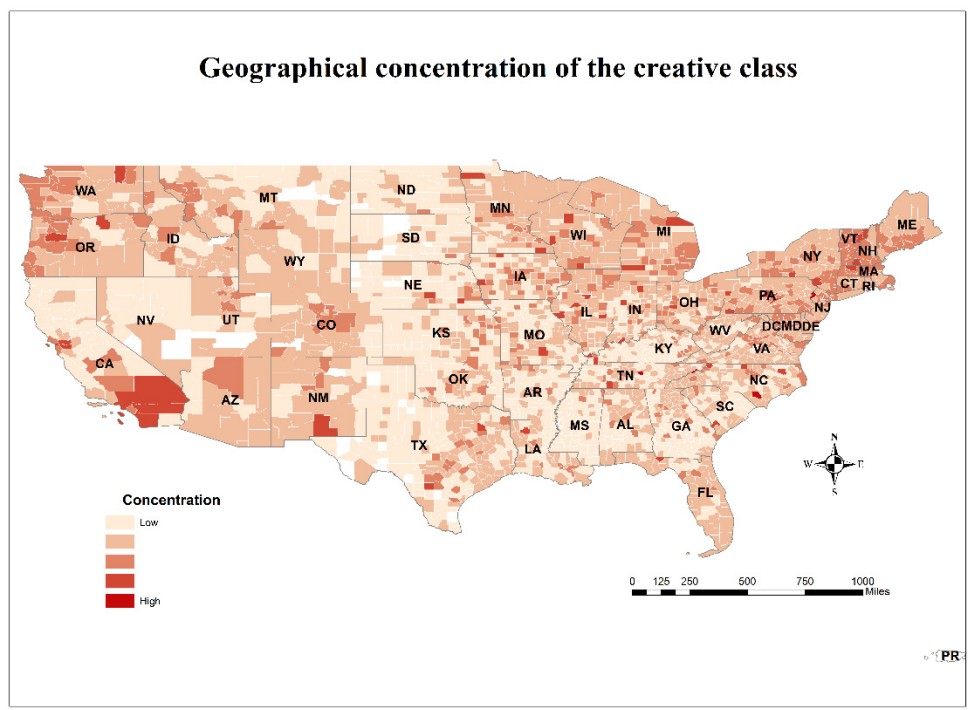

**Figure 2.** The geographical concentration of the creative class. Note: (1) This graph visualizes an average concentration index of all creative class categories collected. (2) Since the absolute magnitude of the index does not have any meaning attached to it, we only show the relative low and high status of different counties in the graph. The five categories of concentrations are formed by equal intervals.

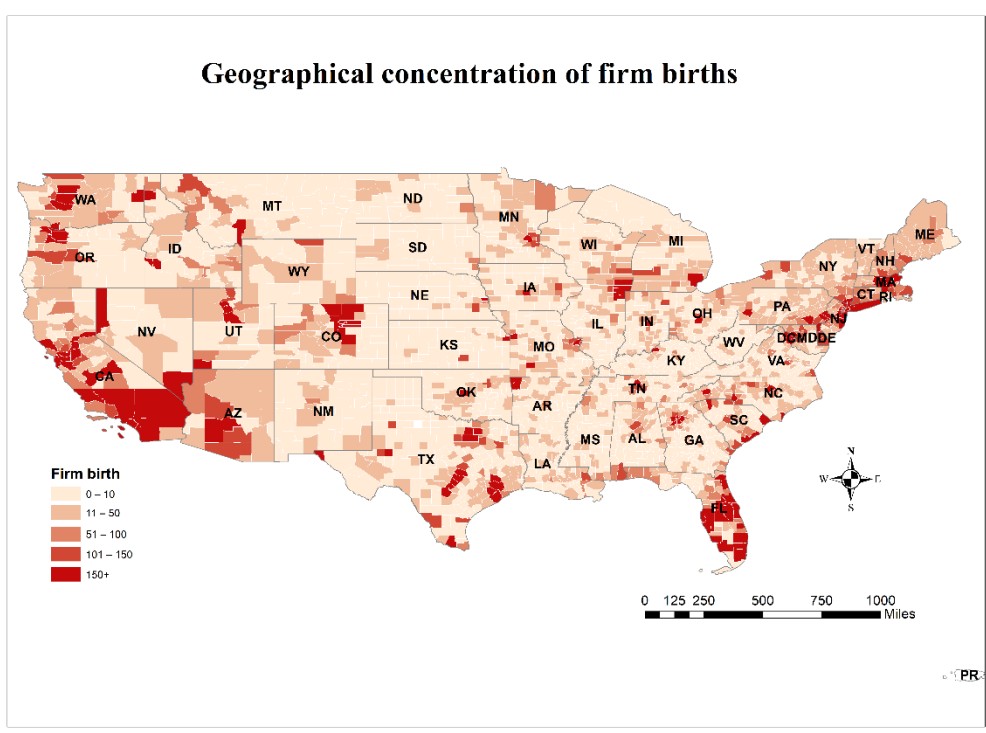

**Figure 3.** The geographical distribution of establishment births in 2018.

### 4.2. The Predictive Power of Urban Third Places

The one hundred models we trained with third place indices and control variables in Table A1 exhibit high explanatory power. The $R^2$ of these 100 models fall between 0.85 to 0.89, with an average of 0.87 (see Figure 4). More than half of the 100 models have an $R^2$

between 0.86 to 0.87. These results show that the trained models are highly effective and also quite stable across different random training samples.

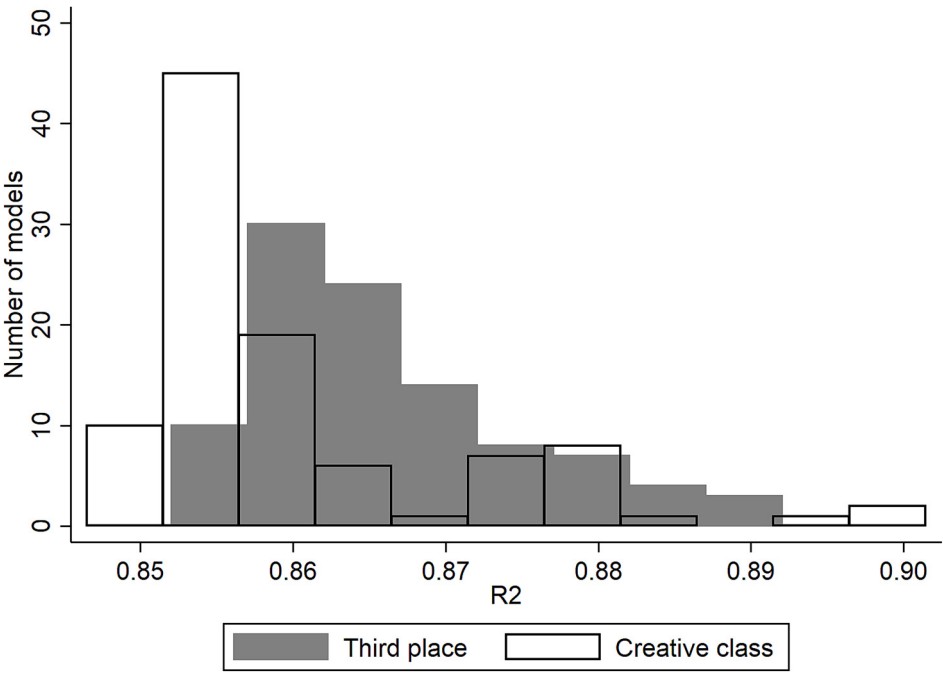

**Figure 4.** The distribution of $R^2$ for trained models.

The predictions made through these models are also quite effective. These 100 trained models, when applying to the test sets containing the other 10% of the samples, yielded an average $R^2$ of 0.72, and a high concentration of $R^2$ between 0.7 to 0.8 (see Figure 5). Of course, the distribution of the $R^2$ in this case is more spread, and there exists an outlier with extremely low predictive power (a negative $R^2$).

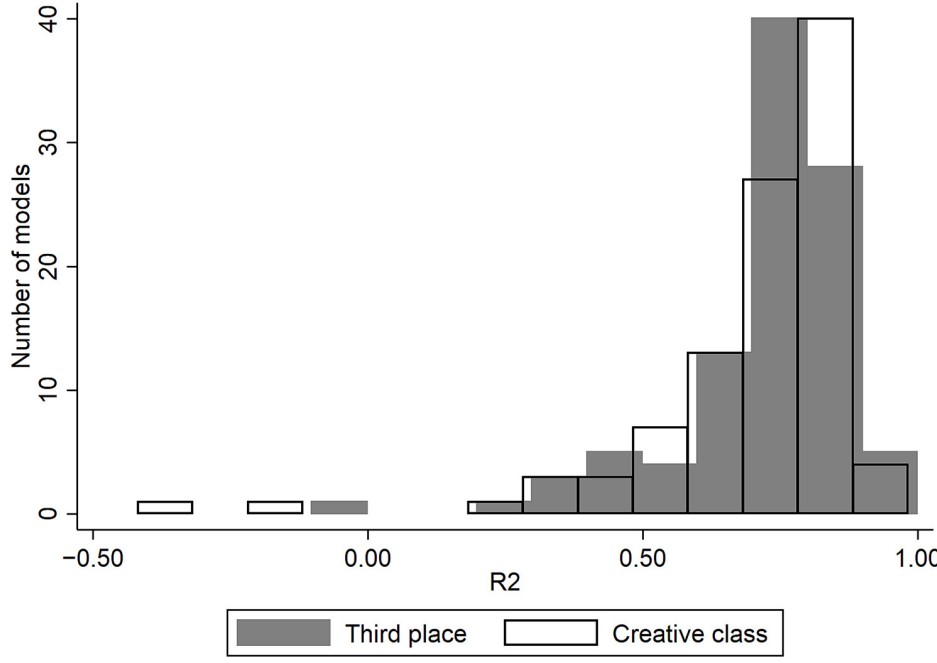

**Figure 5.** The distribution of $R^2$ for predicted models.

Table 1 documents the third-place brands that show the greatest statistical association with entrepreneurship. There are a total of eight brands that are selected into the prediction

models by the Lasso penalty method more than ten times (out of a total of 100 times) as a positive predictor of entrepreneurship. Among these eight brands, three are restaurants—Chick-fil-A, Domino's, and Pizza Hut—out of a total of ten restaurants included in the analysis. Three are coffee shops—The Coffee Bean & Tea Leaf, Peet's Coffee & Tea, and Starbucks—out of a total of five coffee shops in the analysis. Two are bars—Beef 'O' Brady's and Hooters—out of a total of five. In general, coffee shops appear to be the most statistically related to entrepreneurship, compared to restaurants and bars, which echoes the findings of Fang et al. [31]. A higher percentage of coffee shops positively enters the model, and their associated coefficients are on average higher than those of restaurants and bars. Many coffee shops are quieter or sustain quiet spaces that are suitable for business meetings and discussions; this may have contributed to their association to entrepreneurship. The fact that coffee beverages uplift people's spirit and productivity may have also played a role.

**Table 1.** Revealed web browsers' interests in third place brands that are most associated with entrepreneurship.

| Third Place Brand | Number of Models Entered with a Positive Coefficient | Average Coefficient | Standard Deviation of Coefficient |
|---|---|---|---|
| The Coffee Bean & Tea Leaf | 100 | 9.589 | 2.332 |
| Chick-fil-A | 87 | 3.107 | 1.480 |
| Beef 'O' Brady's | 100 | 1.619 | 0.548 |
| Peet's Coffee & Tea | 36 | 0.430 | 0.816 |
| Starbucks | 17 | 0.305 | 0.698 |
| Domino's | 17 | 0.114 | 0.265 |
| Pizza Hut | 15 | 0.072 | 0.190 |
| Hooters | 13 | 0.052 | 0.196 |

Among these eight brands, three stand out. The Coffee Bean & Tea Leaf is selected into the prediction model 100% of the time, and has the largest average coefficient of 9.589, three times as large as the next best predictor, Chick-fil-A. Chick-fil-A, with the second largest average coefficient of 3.107, enters the prediction model 87 times out of a total of 100. It is followed by Beef 'O' Brady's, which is also selected into the prediction model 100% of the time, with an average coefficient of 1.619. These top three brands, interestingly a combination of coffee shops, restaurants and bars, explain most of the variations in entrepreneurship, among all 20 urban third places that we analyzed.

These findings are consistent with prior studies. Manzo and Yilmaz found that since the 1990s, the surge in the number of coffee chains such as the Coffee Bean & Tea Leaf, Starbucks, and Peet's Coffee & Tea has marked the emergence of coffee as a social instrument [54,55]. Coffee houses serve as third places where people engage with each other. Specifically, businessmen and socializing students are among the targeted customers of the Coffee Bean & Tea Leaf. Chick-fil-A has been recognized as a prominent example of the Business as Mission movement [56], which embraces an entrepreneurial model of fostering social responsibility and connection. Millennials and youth activities are the focus of many new Chick-fil-A stores. These unique brand culture may have distinguished them from the other coffee shops and restaurants to become the gathering place for entrepreneurial people and activities.

### 4.3. The Predictive Power of the Creative Class

The creative class indicators also exhibit high explanatory power. However, if compared with the third place indicators, the average explanatory power is slightly lower, and the distribution of $R^2$ is more spread (see Figure 4). Nonetheless, these trained models, when applying to the test set, outperform those with the third place indicators in the majority of cases. Figure 5 shows a high concentration of $R^2$ (about 40% of the models)

between 0.8 to 0.9, compared to the same level of concentration for the third place models falling between 0.7 to 0.8. However, there exist two outliers with an extremely low predictive power and negative $R^2$. These two outliers are worse performers among all models, including third-place models. Given that outliers are rare, only occurring 1.5% of the time among a total of 200 models we trained and tested, the creative class models in most occurrences, we conclude, have a competitive edge over the third-place models in predictive power.

Table 2 summarizes the 15 categories of creative class that are selected into the prediction model by the Lasso penalty method more than ten times, out of a total of 100. These include four types of creative people: (1) the STEM disciplines such as civil engineers and cybersecurity researchers; (2) public policy, conservative think tank researchers and public policy media; (3) those in the arts such as graphic designers, photographers, and painting hobbyists; and (4) writers and those in allied interests such as writing and publishing.

**Table 2.** Creative class categories revealed by web browsers' interests that are most associated with entrepreneurship.

| Creative Class Category | Number of Models Entered with a Positive Coefficient | Average Coefficient | Standard Deviation of Coefficient |
|---|---|---|---|
| Civil Engineers | 94 | 2.991 | 0.960 |
| Cybersecurity Researchers | 98 | 1.691 | 0.812 |
| Conservative Think Tank Researchers | 96 | 1.092 | 0.453 |
| Design Software | 99 | 0.859 | 0.327 |
| Writing & Publishing | 97 | 0.479 | 0.245 |
| Graphic Design | 97 | 0.414 | 0.243 |
| Public Policy Media | 84 | 0.342 | 0.248 |
| Photography Interest | 24 | 0.228 | 0.500 |
| Data Analysis and Scripting | 41 | 0.172 | 0.312 |
| Science & Technology | 20 | 0.149 | 0.425 |
| Film Production | 60 | 0.133 | 0.171 |
| Memes & Comedy | 42 | 0.067 | 0.122 |
| Science Education & Academic | 10 | 0.017 | 0.063 |
| Painting Hobbyists | 14 | 0.015 | 0.111 |

Among these 15 categories, four stand out with a coefficient larger than 0.5. Three of them are high-tech related: (1) civil engineers, with an average coefficient of 2.991; (2) cybersecurity researchers, with an average coefficient of 1.691; and (3) design software, with an average coefficient of 0.859. Conservative think tank researchers are another category that is statistically associated with establishment births with a large coefficient—an average coefficient of 1.092. These findings are consistent with prior research showing that engineering and computer science, as well as other STEM fields, account for the majority of entrepreneurship in the US [57,58]. However, we identified think tank researchers as another group of creative class that is correlated with entrepreneurship.

### 4.4. Robustness Checks with Metropolitan versus Non-Metropolitan Counties

Table 3 tests whether the rate of prediction differs for metropolitan and non-metropolitan counties. Metropolitan and non-metropolitan counties are defined following the USDA's rural urban continuum 2013 definition. As mentioned above, we expect the prediction rate to be lower in non-metropolitan counties because of the more sporadic distribution of the access to internet in those counties. Table 3 confirms our expectations. For the training sets, the average $R^2$ for third place indicators in metropolitan counties is 0.867, compared to only

0.479 in non-metropolitan counties. Similarly, for the creative class indicators, the average $R^2$ for metropolitan counties is 0.861, and that for non-metropolitan counties is only 0.478.

**Table 3.** Explained variation in entrepreneurship for metropolitan versus non-metropolitan counties.

| | Third Places | | Creative Class | |
|---|---|---|---|---|
| | **Metro** | **Non-Metro** | **Metro** | **Non-Metro** |
| Training set average $R^2$ | 0.867 | 0.479 | 0.861 | 0.478 |
| Training set $R^2$ standard deviation | 0.008 | 0.011 | 0.007 | 0.020 |
| Testing set average $R^2$ | 0.682 | 0.365 | 0.687 | 0.378 |
| Testing set $R^2$ standard deviation | 0.230 | 0.107 | 0.174 | 0.102 |

The prediction rates remain drastically different in the testing sets. The average $R^2$ is 0.682 and 0.687 for metropolitan counties for third place and creative class indicators, respectively. In comparison, the average $R^2$ for non-metropolitan counties is only about half as much, at 0.365 and 0.378, respectively. These findings suggest that web-browsing behavior is a better predictor for entrepreneurship in urban areas than in rural places, but it may also indicate that rural areas also have lower rates and quality of broadband access.

Moreover, in both the metropolitan and non-metropolitan testing sets, the average $R^2$ for the creative class indicators outperform that of the third place indicators. The competitive advantage of the creative class indicators is larger for non-metropolitan counties where statistically explaining entrepreneurship is difficult. Overall, this is consistent with the findings in Section 4.3, confirming that the creative class indicators are better predictors.

## 5. Discussion

Our results consistently show that while both the third places and the creative class are associated with entrepreneurship, the creative class exhibit a small advantage. This indicates that in encouraging entrepreneurship and attracting and cultivating human capital is slightly more important than the "quality of place". That said, both people-based and place-based approaches are justifiable. Great spaces are important in sustaining social interaction, but they are better filled with creative people. Not all social interactions lead to innovation and entrepreneurship, but with the creative class gathering in well-designed places, fresh ideas and startup businesses are more likely to be born. As a result, urban planners who adopt both approaches and prioritize human capital may be more successful in encouraging an entrepreneurial business environment in their jurisdictions.

We have also explored the boundary for such conclusions. Our results show that either third places or the creative class are strongly related with entrepreneurship in metropolitan counties but do not perform as well in non-metropolitan counties. Thus, methodologically, using web browsing behaviors to predict entrepreneurship in non-metropolitan counties may only be relevant in a comparative sense across regions. A heat map of creative class or third place web-browsing behaviors over a couple months may be a relative indicator of coming business formation. Over time, one can see the degree to which these cultural differences across space can forecast differences in business activities.

This paper is among the first to identify which third places and which creative class categories are conducive to entrepreneurship. Academically, narrowing down this list helps scholars to further unpack the underlying mechanisms that propel entrepreneurship. For example, we found coffee shops to be a complement to entrepreneurship; this finding helps further our understanding of the specific type of spaces that can better support business-related conversations and potentially lead to establishment births. Similarly, for the creative class, we found that the STEM- and IT-related creative class drives entrepreneurship more than art-related ones. This shows how STEM, and specifically IT, are still the main forces driving business formation in the United States. Moreover, the identification of specific third place brands and creative class categories also point to a direction for future research, especially for in-depth case studies. For example, the Coffee Bean & Tea Leaf is the most powerful predictor of entrepreneurship across all studied third-place brands. Is there

something unique about this brand? Is it unconventional while other coffee brands are more geographically saturated and therefore more pedestrian? Similarly, is there a possible explanation why civil engineers, more than any other creative class categories, are more closely associated with entrepreneurship? One may have hypothesized architects and artistic designers.

Narrowing down the list of third places and the creative class categories also has important practical implications. These specific types of third places or creative class identified can assist policymakers in properly designing entrepreneurship policies. For example, it may be more fiscally responsible to fund and sponsor training programs that develop certain types of creativity, such as collaborative educational programs between STEM departments in universities and business incubators, or events that engage hobbyists in data science, rather than larger physical, and irreversible, infrastructure projects that require a ten- to twenty-year bond commitment.

## 6. Conclusions

Using contemporaneous and geographically granular user online activity data, we subject two theories to competition: urban third places and the creative class. In this way, together with machine learning methods, we assess which theory explains more of entrepreneurial activities at the county level in the United States. Both exhibit high explanatory power and explained variations of more than 70%. The creative class has a slight competitive advantage in statistical association than third places.

This paper makes theoretical, practical, and methodological contributions. Theoretically, it supports the power and relevance of both urban third places and the creative class theories. Our empirical evidence shows that the creative class theory has a slight competitive edge and all things equal, cultivating human capital wins on the margin. Thus, practically, both place-based and people-based policy initiatives are supported. If budgetary considerations were not constrained, this paper shows that people-based initiatives that cultivate the creative class should be prioritized.

Methodologically, this paper uses nearly real-time data with the potential to nowcast entrepreneurship, at least to identify comparatively hot spots for entrepreneurship across regions. The high prediction rate (more than 70%) testifies to the effectiveness of this approach. Moreover, this paper improves the measurement of the creative class. Prior studies have commonly identified the creative class using the occupation data. Creativity, however, may or may not be tied to professional occupations—paper-hanger by day but saxophonist by night. This paper solves this problem using web browsing behaviors that capture user revealed interests and avocational aspirations. We posit that this is a better approach to measure a broader array of characteristics of the creative class.

However, this paper and the data it employs also have two major limitations. First, web user-generated data are more representative in urban areas. Thus, the analysis for rural counties is less accurate. In this study, we have tested the prediction power for urban and rural counties, and that for the latter is only about half as robust compared to the former. Second, the data collected for urban third places and the creative class are admittedly incomplete. For the top ten coffee shops and bars by sales, only five are available in the Dstillery dataset. This is because Dstillery, Inc. captures behavior on the web and tracks it for their clients to post personally targeted advertisements on websites the users browse. Since not all brands are Dstillery clients, the profiles for some brands are not available. Similarly, not all creative class types are included in the Dstillery dataset. Nevertheless, our results show that the prediction rates for urban third places and the creative class are both quite high—more than 70%. Thus, we believe, though imperfect, these are trustworthy measurements.

**Author Contributions:** Conceptualization, L.F. and T.S.; methodology, L.F.; software, L.F.; formal analysis, L.F.; resources, T.S.; data curation, L.F. and T.S.; writing—original draft preparation, L.F.; writing—review and editing, L.F. and T.S.; visualization, L.F. All authors have read and agreed to the published version of the manuscript.

**Funding:** This research received no external funding.

**Institutional Review Board Statement:** Not applicable.

**Informed Consent Statement:** Not applicable.

**Data Availability Statement:** Data are provided by Dstillery, Inc. Researchers who are interested in learning more about the Dstillery data, please contact Timothy Slaper at tslaper@indiana.edu.

**Acknowledgments:** The authors thank our outstanding research assistant Yijia Wen (Florida State University) for programming assistance.

**Conflicts of Interest:** The authors declare no conflict of interest.

## Appendix A

**Table A1.** Variable description.

| Variable | Mean | Standard Deviation | Min | Max | Observations | Data Source |
|---|---|---|---|---|---|---|
| Dependent variable: | | | | | | |
| Establishment birth | 35.378 | 155.078 | 0 | 6002 | 3074 | Census Bureau Business Dynamics Statistics, 2018 |
| Explanatory variables: | | | | | | |
| Concentration index for urban third place brands | | | | | | |
| Beef 'O' Brady's | 0.790 | 2.377 | 0 | 32.447 | 3077 | Dstillery, Inc., 2020 |
| Buffalo Wild Wings | 0.891 | 0.502 | 0 | 4.762 | 3077 | Dstillery, Inc., 2020 |
| Burger King | 0.977 | 0.342 | 0 | 3.037 | 3077 | Dstillery, Inc., 2020 |
| Caribou Coffee | 1.466 | 3.052 | 0 | 28.044 | 3077 | Dstillery, Inc., 2020 |
| Chick-fil-A | 0.816 | 0.521 | 0 | 3.909 | 3077 | Dstillery, Inc., 2020 |
| Chili's Grill & Bar | 0.807 | 0.586 | 0 | 3.765 | 3077 | Dstillery, Inc., 2020 |
| Chipotle Mexican Grill | 0.664 | 0.506 | 0 | 3.974 | 3077 | Dstillery, Inc., 2020 |
| The Coffee Bean & Tea Leaf | 0.638 | 1.754 | 0 | 37.429 | 3077 | Dstillery, Inc., 2020 |
| Dave & Buster's | 0.558 | 0.719 | 0 | 9.088 | 3077 | Dstillery, Inc., 2020 |
| Domino's | 0.909 | 0.393 | 0 | 3.499 | 3077 | Dstillery, Inc., 2020 |
| Dunkin Donuts | 0.672 | 0.514 | 0 | 4.026 | 3077 | Dstillery, Inc., 2020 |
| Hooters | 0.694 | 0.743 | 0 | 6.878 | 3077 | Dstillery, Inc., 2020 |
| McDonald's | 1.014 | 0.291 | 0.292 | 2.875 | 3077 | Dstillery, Inc., 2020 |
| Panera Bread | 0.717 | 0.451 | 0 | 3.411 | 3077 | Dstillery, Inc., 2020 |
| Peet's Coffee & Tea | 0.690 | 3.119 | 0 | 47.697 | 3077 | Dstillery, Inc., 2020 |
| Pizza Hut | 1.065 | 0.386 | 0 | 3.429 | 3077 | Dstillery, Inc., 2020 |
| Starbucks | 0.867 | 0.352 | 0.195 | 2.782 | 3077 | Dstillery, Inc., 2020 |
| Subway | 1.056 | 0.289 | 0.304 | 2.533 | 3077 | Dstillery, Inc., 2020 |
| Taco Bell | 0.983 | 0.352 | 0 | 3.562 | 3077 | Dstillery, Inc., 2020 |
| Wendy's | 0.924 | 0.356 | 0 | 2.584 | 3077 | Dstillery, Inc., 2020 |
| Concentration index for different types of the creative class | | | | | | |
| Architects | 0.844 | 0.341 | 0 | 3.256 | 3074 | Dstillery, Inc., 2020 |
| Art News & Products | 0.910 | 1.157 | 0 | 19.228 | 3074 | Dstillery, Inc., 2020 |
| Arts & Crafts | 1.368 | 2.015 | 0 | 32.877 | 3074 | Dstillery, Inc., 2020 |
| Authors | 0.914 | 0.180 | 0 | 2.077 | 3074 | Dstillery, Inc., 2020 |
| Civil Engineers | 1.000 | 0.190 | 0 | 2.083 | 3074 | Dstillery, Inc., 2020 |
| College Professors | 0.843 | 0.361 | 0 | 3.913 | 3074 | Dstillery, Inc., 2020 |
| Commercial Architects | 0.841 | 0.215 | 0 | 2.281 | 3074 | Dstillery, Inc., 2020 |
| Commercial Contractors | 0.843 | 0.202 | 0 | 1.982 | 3074 | Dstillery, Inc., 2020 |
| Conservative Think Tank Researchers | 0.987 | 0.179 | 0 | 2.264 | 3074 | Dstillery, Inc., 2020 |
| Cybersecurity Researchers | 0.866 | 0.193 | 0 | 2.152 | 3074 | Dstillery, Inc., 2020 |
| Data Analysis and Scripting | 0.650 | 0.469 | 0 | 4.655 | 3074 | Dstillery, Inc., 2020 |
| Design Software | 0.742 | 0.511 | 0 | 5.856 | 3074 | Dstillery, Inc., 2020 |
| Drawing & Animation | 3.044 | 11.596 | 0 | 288.001 | 3074 | Dstillery, Inc., 2020 |

**Table A1.** *Cont.*

| Variable | Mean | Standard Deviation | Min | Max | Observations | Data Source |
|---|---|---|---|---|---|---|
| Drawing Enthusiasts | 0.940 | 0.251 | 0 | 2.328 | 3074 | Dstillery, Inc., 2020 |
| Entertainment Industry Decision Makers | 0.801 | 0.257 | 0 | 2.883 | 3074 | Dstillery, Inc., 2020 |
| Film Production | 0.874 | 1.016 | 0 | 18.359 | 3074 | Dstillery, Inc., 2020 |
| Graphic Design | 0.871 | 0.351 | 0 | 3.293 | 3074 | Dstillery, Inc., 2020 |
| Healthcare Thought Leaders | 0.871 | 0.209 | 0 | 3.258 | 3074 | Dstillery, Inc., 2020 |
| Humor & Entertainment-Comic Culture | 0.907 | 0.237 | 0 | 2.417 | 3074 | Dstillery, Inc., 2020 |
| Landscape Architects & Designers | 0.884 | 0.181 | 0.163 | 2.113 | 3074 | Dstillery, Inc., 2020 |
| Liberal Think Tank Researchers | 0.862 | 0.199 | 0 | 1.85 | 3074 | Dstillery, Inc., 2020 |
| Live Music | 0.870 | 0.347 | 0 | 2.913 | 3074 | Dstillery, Inc., 2020 |
| Medical Science Researchers | 0.848 | 0.291 | 0 | 2.638 | 3074 | Dstillery, Inc., 2020 |
| Memes & Comedy | 2.248 | 5.444 | 0 | 102.719 | 3074 | Dstillery, Inc., 2020 |
| Music Concerts | 0.912 | 0.274 | 0 | 2.323 | 3074 | Dstillery, Inc., 2020 |
| Musical Instrument Purchasers | 0.897 | 0.181 | 0 | 2.303 | 3074 | Dstillery, Inc., 2020 |
| Painting & Renovation | 0.869 | 0.611 | 0 | 5.369 | 3074 | Dstillery, Inc., 2020 |
| Painting Hobbyists | 0.820 | 0.747 | 0 | 5.63 | 3074 | Dstillery, Inc., 2020 |
| Photography Interest | 0.925 | 0.186 | 0 | 2.458 | 3074 | Dstillery, Inc., 2020 |
| Poetry Fans | 0.819 | 0.296 | 0 | 4.770 | 3074 | Dstillery, Inc., 2020 |
| Public Policy Media | 0.864 | 0.481 | 0 | 4.832 | 3074 | Dstillery, Inc., 2020 |
| Poetry Readers | 0.941 | 0.293 | 0 | 3.516 | 3074 | Dstillery, Inc., 2020 |
| Science & Technology | 0.902 | 0.209 | 0 | 2.698 | 3074 | Dstillery, Inc., 2020 |
| Science Education & Academic | 0.923 | 0.797 | 0 | 8.338 | 3074 | Dstillery, Inc., 2020 |
| University Research | 0.828 | 0.319 | 0 | 3.620 | 3074 | Dstillery, Inc., 2020 |
| Writing & Publishing | 0.932 | 0.886 | 0 | 12.832 | 3074 | Dstillery, Inc., 2020 |
| Writing Tools & Citation | 0.813 | 0.382 | 0 | 7.834 | 3074 | Dstillery, Inc., 2020 |
| Control variables: | | | | | | |
| County 2010 census population | 189,592.4 | 1,161,204 | 82 | $3.73 \times 10^{-7}$ | 3271 | Census Bureau |
| County 2019 population, estimated | 201,150.5 | 1,243,844 | 86 | $3.95 \times 10^{-7}$ | 3271 | Census Bureau |
| County 2010 population density (persons/miles) | 262.839 | 1744.994 | 0.1 | 70,148.7 | 3143 | Census Bureau |
| Percent of female, 2010 | 0.500 | 0.022 | 0.279 | 0.568 | 3142 | Census Bureau |
| Percent of white alone population, 2010 | 0.858 | 0.164 | 0.027 | 0.997 | 3142 | Census Bureau |
| Percent of Black or African American alone population, 2010 | 0.090 | 0.146 | 0 | 0.857 | 3142 | Census Bureau |
| Percent of American Indian or Alaska Native alone population, 2010 | 0.022 | 0.078 | 0 | 0.961 | 3142 | Census Bureau |
| Percent of Asian alone population, 010 | 0.012 | 0.026 | 0 | 0.444 | 3142 | Census Bureau |
| Percent of Native Hawaiian and Pacific Islander alone population, 2010 | 0.001 | 0.010 | 0 | 0.489 | 3142 | Census Bureau |
| Percent of two or more race population, 2010 | 0.017 | 0.015 | 0 | 0.291 | 3142 | Census Bureau |
| Percent of age 0–4, 2010 | 0.063 | 0.012 | 0 | 0.126 | 3142 | Census Bureau |
| Percent of age 5–9, 2010 | 0.064 | 0.01 | 0 | 0.121 | 3142 | Census Bureau |
| Percent of age 10–14, 2010 | 0.066 | 0.009 | 0 | 0.121 | 3142 | Census Bureau |
| Percent of age 15–19, 2010 | 0.069 | 0.012 | 0 | 0.183 | 3142 | Census Bureau |
| Percent of age 20–24, 2010 | 0.060 | 0.026 | 0.013 | 0.330 | 3142 | Census Bureau |
| Percent of age 25–29, 2010 | 0.058 | 0.012 | 0.023 | 0.161 | 3142 | Census Bureau |

**Table A1.** *Cont.*

| Variable | Mean | Standard Deviation | Min | Max | Observations | Data Source |
|---|---|---|---|---|---|---|
| Percent of age 30–34, 2010 | 0.057 | 0.010 | 0.024 | 0.117 | 3142 | Census Bureau |
| Percent of age 35–39, 2010 | 0.059 | 0.009 | 0.012 | 0.097 | 3142 | Census Bureau |
| Percent of age 40–44, 2010 | 0.063 | 0.009 | 0.028 | 0.119 | 3142 | Census Bureau |
| Percent of age 45–49, 2010 | 0.074 | 0.008 | 0.033 | 0.137 | 3142 | Census Bureau |
| Percent of age 50–54, 2010 | 0.076 | 0.008 | 0.027 | 0.129 | 3142 | Census Bureau |
| Percent of age 55–59, 2010 | 0.070 | 0.011 | 0.019 | 0.189 | 3142 | Census Bureau |
| Percent of age 60–64, 2010 | 0.062 | 0.012 | 0.021 | 0.130 | 3142 | Census Bureau |
| Percent of age 65–69, 2010 | 0.049 | 0.012 | 0.012 | 0.158 | 3142 | Census Bureau |
| Percent of age 70–74, 2010 | 0.038 | 0.010 | 0.008 | 0.128 | 3142 | Census Bureau |
| Percent of age 75–79, 2010 | 0.029 | 0.009 | 0.003 | 0.1 | 3142 | Census Bureau |
| Percent of age 80–84, 2010 | 0.022 | 0.007 | 0.003 | 0.1 | 3142 | Census Bureau |
| Percent of age 85 and above, 2010 | 0.021 | 0.009 | 0 | 0.083 | 3142 | Census Bureau |
| Percent of adults with less than a high school diploma, 2014–2018 | 13.700 | 6.646 | 1.2 | 66.3 | 3273 | American Community Survey |
| Percent of adults with a high school diploma only, 2014–2018 | 34.085 | 7.175 | 5.5 | 55.6 | 3273 | American Community Survey |
| Percent of adults completing some college or associate's degree, 2014–2018 | 30.493 | 5.347 | 5.8 | 57.3 | 3273 | American Community Survey |
| Percent of adults with a bachelor's degree or higher, 2014–2018 | 21.720 | 9.397 | 0 | 78.5 | 3273 | American Community Survey |
| Unemployment rate 2019 | 3.965 | 1.380 | 1.4 | 18.3 | 3069 | Bureau of Labor Statistics |
| Median household income 2018 | 52,796.7 | 13,853.84 | 25,385 | 140,382 | 3069 | Census Bureau |
| Population in poverty 2018 | 13,615.7 | 46,094.12 | 65 | 1,409,155 | 3069 | Census Bureau |
| Rural-urban continuum code 2003 (1–9) | 4.938 | 2.725 | 1 | 9 | 3219 | U.S. Department of Agriculture |
| Urban influence code 2003 (1–12) | 5.190 | 3.506 | 1 | 12 | 3219 | U.S. Department of Agriculture |

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
