# Peer review of "Nowcasting Entrepreneurship: Urban Third Place versus the Creative Class"

_sustainability, doi:10.3390/su14020763_

Round 1

Reviewer 1 Report

The methodology used in the work does not clarify the way in which the research objectives will be achieved.

The paper addresses in a somewhat diffuse manner the objective pursued and the main contributions derived from the results.

In the paper, more than 54% of the references are older than 5 years.

Author Response

Dear Reviewer 1, Thank you very much for your suggestion to our paper. We’ve listed our responses to your comments below, with revisions in the main text marked in track changes.

  1. The methodology used in the work does not clarify the way in which the research objectives will be achieved.

We have revised the methodology section (section 3.2) to state how our method achieves the research objective.

  1. The paper addresses in a somewhat diffuse manner the objective pursued and the main contributions derived from the results.

We’ve tightened up the research objective, contribution and the conclusion sections in the revised version to better focus and align these items in sections 1 and 6. We highlighted our main research objective (comparing the explanatory power of the third place versus the creative class of regional entrepreneurial activities), and summarize our contribution beginning with findings associated with this objective, and its practical/policy implications, and then add on our methodological contributions. This new adjustment flows better and logically tightens up.

  1. In the paper, more than 54% of the references are older than 5 years.

We’ve revisited the reference list, dropped some older references and updated with some newer ones.

Reviewer 2 Report

The statement line 40-41 needs a reference, because is a new, very important concept. Actually the place and people are in a kind of interaction, building everyday atmosphere and opportunities for creative thinking (compare the idea of pedagogy of places).

When the authors are writing about social capital, it needs to be mentioned specific understanding of creative and critical thinking (one paragraph for each topic will be enough), which are characteristics of creative class (2.3)

Research limitations (line 252)  should be mentioned in a separate part of the manuscript, at the end of

Author Response

Dear Reviewer 2, Thank you very much for your suggestion for our paper. We’ve listed our responses to your comments below, with revisions in the main text marked in track changes.

  1. The statement line 40-41 needs a reference, because is a new, very important concept. Actually the place and people are in a kind of interaction, building everyday atmosphere and opportunities for creative thinking (compare the idea of pedagogy of places).

Thank you for your suggestion. We’ve added citations to two relevant articles.

  1. When the authors are writing about social capital, it needs to be mentioned specific understanding of creative and critical thinking (one paragraph for each topic will be enough), which are characteristics of creative class (2.3)

Thank you for your suggestion. We’ve expanded these discussions in section 2.3.

  1. Research limitations (line 252) should be mentioned in a separate part of the manuscript, at the end of the Conclusions.

Thank you. We moved them to the end of the Conclusions section in the revised version.